

**Improvement of typhoon wind hazard model and its sensitivity**
**analysis**
Yunxia Guo[1,2,4], Yijun Hou [1,3,4*], and Peng Qi [1,3,4*]
[1] Key Laboratory of Ocean Circulation and Waves, Institute of Oceanology, Chinese Academy of Sciences, Qingdao, 7 Nanhai Road,
266071, China;
[2] University of Chinese Academy of Sciences, Beijing, 19A Yuquan Road, 100049 China;
[3] Laboratory for Ocean and Climate Dynamics, Qingdao National Laboratory for Marine Science and Technology, Qingdao, 1 Wenhai
Road, 266237, China;
[4] Center for Ocean Mega-Science, Chinese Academy of Sciences, Qingdao, 7 Nanhai Road, 266071, P. R. China
* Corresponding author.
E-mail address: guoyunxia14@mails.ucas.edu.cn (Yunxia Guo), yjhou@qdio.ac.cn (Yijun Hou), pqi@qdio.ac.cn (Peng Qi).
**Abstract.** Typhoons are one of the most serious natural disasters that occur annually on China's
southeast coast. This paper describes a technique for analyzing the typhoon wind hazard based on
the empirical track model. Existing simplified and non-simplified typhoon empirical track models
are improved, and the improved tracking models are shown to significantly increase the
correlation in regression analysis. We also investigate quantitatively the sensitivity of the typhoon
wind hazard model. The effects of different typhoon decay models, the simplified and
non-simplified typhoon tracking models, different statistical models for the radius to maximum
winds ($R_{max}$) and Holland pressure profile parameter ($B$), and different extreme value distributions
on the predicted extreme wind speed of different return periods are all investigated. Comparisons
of estimated typhoon wind speeds for 50-year and 100-year return periods under the influence of
different factors are presented. The different models of $R_{max}$ and $B$ are found to have greatest
impact on the prediction of extreme wind speed, followed by the extreme value distributions,
typhoon tracking models, and typhoon decay models. This paper constitutes a useful reference for
predicting extreme wind speed using the empirical track model.

**Keywords**
empirical track model; decay model; sensitivity; extreme value distribution; extreme wind speed.

**1 Introduction**
China's southeast coast is the region of the world that suffers most from severe typhoon
disasters. Typhoons, known as hurricanes in the eastern Pacific and Atlantic oceans, can create
complex environments of high winds, heavy rainfall, huge wave heights, and huge storm surges
throughout the region. Therefore, it is very important to analyze the typhoon hazard using typhoon
wind hazard modeling and simulation methods.
In the second half of the 20th century, the Monte Carlo simulation was adopted most widely





for performing typhoon hazard analysis. It uses a mature typhoon model and typhoon history data to simulate the typhoon wind field and to predict the annual maximum wind speed. Both the United States of America (ASCE/SEI 7-05) and Australia (SAA, 2002) use the method to compile design wind speed maps.

The simulation approach was first implemented by Russell (1969, 1971) for the Texas coast (USA). Since that pioneering study, the modeling technique has been expanded and improved by Batts et al. (1980), Shapiro (1983), Georgiou et al. (1983), Vickery and Twisdale (1995b), Meng et al. (1995), Simiu and Scanlan (1996), and Thompson and Cardone (1996). As indicated by Vickery and Twisdale (1995a), although the approaches used by these investigators are similar, there are significant differences in the decay models, wind field models, size of the region over which the typhoon climatology can be considered uniform, and use of a coast segment crossing approach.

Since 2000, the full-track modeling method has gradually been developed (Vickery et al., 2000, 2009b; Huang et al., 2001; James and Mason 2005; Emanuel, 2006; Emanuel et al., 2006; Hall and Jewson 2007). Vickery et al. (2000) were pioneers of full-track modeling and they developed an empirical track model. This model can generate the full track of a typhoon from generation to extinction. As indicated by Vickery et al. (2000), an improvement of the storm track modeling approach over a Monte Carlo simulation is that it is not dependent on the hypothesis of climate uniformity in the subregion. Therefore, even in a large region with considerable change in typhoon climatology, it remains appropriate for typhoon hazard analysis, which is helpful for analyzing the hazard of large-scale systems. The empirical track model has been used in many studies for typhoon hazard analysis (Powell et al., 2005; Lee and Rosowsky, 2007; Legg et al., 2010; Apivatanagul et al., 2011; Pei et al., 2014; Li and Hong, 2015b, 2016). The design wind speeds recommended by U.S. building codes (ASCE 7-10, 2010) are also based on the empirical track model (Vickery et al. 2000).

The process of analyzing typhoon hazard using the empirical track model is that first a large number of virtual typhoons is generated using the typhoon empirical track model and the decay model. Then, the typhoons that affect a certain research site are extracted from the virtual typhoons using the simulated circle method. Next, a typhoon wind field model is used to calculate the wind speed of the extracted typhoons, from which samples of maximum wind speed can be derived. Finally, the samples of maximum wind speed are fitted by some extreme value distribution, based on which extreme wind speeds for different return periods can be predicted. Many factors can influence the prediction of extreme wind speed throughout the entire process. The empirical track model developed by Vickery et al. (2000) has been simplified by Li and Hong (2015b) through the adoption of the geographic weighted regression method (Fotheringham et al.


2002), and they also fully validated the efficiency of the simplified tracking model. Subsequently,
Vickery and Wadhera (2008) and Vickery et al. (2009a) updated the statistical model for the
radius to maximum winds ($R_{max}$) and the Holland pressure profile parameter ($B$) using pressure
data from hurricane reconnaissance flights and information of hurricane wind fields from the
Hurricane Research Division's H*Wind snapshots. Vickery (2005) also developed a new model
for hurricane decay after landfall. It was found that the hurricane decay rate is correlated
positively (negatively) with the central pressure difference and translation speed at the time of
landing ($R_{max}$) along the coasts of the Gulf of Mexico and the Florida Peninsula. However, along
the Atlantic coast, it was found that $R_{max}$ has minimal importance in the hurricane decay rate.
This paper investigates the typhoon wind hazard model from two perspectives. The first is
the improvement of the typhoon tracking models consisting of the simplified and non-simplified
models. We find the improved tracking models can significantly increase the correlation in
regression analysis. The second aspect is the sensitivity of the typhoon wind hazard model to
different influencing factors including different typhoon decay models, the simplified and
non-simplified typhoon tracking models, different statistical models for $R_{max}$ and $B$, and different
extreme value distributions. The effects of these factors on predicted extreme wind speed for
50-year and 100-year return periods in the southeast coastal region of China are investigated
quantitatively. This work constitutes a useful reference for predicting extreme wind speed using an
empirical track model.

## 2 Empirical track models

Vickery et al. (2000) developed the typhoon empirical track model, which models the
typhoon translation speed, storm heading, and relative intensity. The model is expressed as:
$$\Delta \ln c = a_1 + a_2 \psi + a_3 \lambda + a_4 \ln c_i + a_5 \theta_i + \varepsilon_c , \tag{1a}$$

$$\Delta \theta = b_1 + b_2 \psi + b_3 \lambda + b_4 c_i + b_5 \theta_i + b_6 \theta_{i-1} + \varepsilon_\theta , \tag{1b}$$

$$\ln(I_{i+1}) = d_1 + d_2 \ln(I_i) + d_3 \ln(I_{i-1}) + d_4 \ln(I_{i-2}) + d_5 Ts_i + d_6(Ts_{i+1} - Ts_i) + \varepsilon_I , \tag{1c}$$

where coefficients $a_i$, $b_i$, and $d_i$ are developed on a $5\,^\circ \times 5\,^\circ$ grid over the entire Northwest Pacific
Basin, based on regression analysis of historical typhoon data; $\psi$ and $\lambda$ represent the storm latitude
(°) and longitude (°), respectively; $c_i$, $\theta_i$, and $I_i$ are the typhoon translation speed, storm heading,
and relative intensity, respectively, at time step of $i$; $\Delta \ln c = \ln c_{i+1} - \ln c_i$; $\Delta \theta = \theta_{i+1} - \theta_i$; $T_{si}$ is
monthly mean sea surface temperature (K); and $\varepsilon_c$, $\varepsilon_\theta$, and $\varepsilon_I$ are random error terms. The historical
typhoon dataset used here is the China Meteorological Administration–Shanghai Typhoon
Institute Best Track Dataset for Tropical Cyclones over the Western North Pacific (1949–2017,
from www.typhoon.gov.cn).





The relative intensity $I$ is defined as (Darling, 1991):
$$I = \Delta p / (p_{da} - p_{dc}),\qquad\qquad(2)$$
where $p_{da}$ and $p_{dc}$ are the ambient and minimum sustainable central dry partial pressures,
respectively, and $\Delta p$ is the central pressure difference. For details on the specific method for the
calculation of relative intensity, the reader is referred to Darling (1991). We distinguish easterly
and westerly headed storms, and we obtain two set of coefficients ($a_i$, $b_i$, and $d_i$) for both types.
When a grid cell has few or no historical typhoons, the coefficients are replaced with those of the
nearest grid cell.
In the tracking model of Vickery et al. (2000), many coefficients have to be determined for
each grid cell. Li and Hong (2015b) eliminated some secondary explanatory variables in the
regression model and they simplified the tracking model of Vickery et al. (2000) using the
geographic weighted regression method (Fotheringham et al. 2002). The simplified tracking
model can be expressed as follows:
$$\Delta \ln c = a_1 + a_2 \ln c_i + a_3 \theta_i + \varepsilon_c,\qquad\qquad(3a)$$
$$\Delta \theta = b_1 + b_2 c_i + b_3 \theta_i + \varepsilon_\theta,\qquad\qquad(3b)$$
$$\ln(I_{i+1}) = d_1 + d_2 \ln(I_i) + d_3 Ts_i + d_4 (Ts_{i+1} - Ts_i) + \varepsilon_I.\quad(3c)$$
Li and Hong (2015b) compared the standard deviations of the residuals in the regression analysis
for Eqs. (1) and (3) and they indicated that the fit obtained by Eq. (3) is comparable with Eq. (1).
To further validate the simplified tracking model, they also compared the statistics of typhoons
simulated using the simplified model with observed data and they found the simplified model
efficient.

## 2.1 Improvement of the empirical track model

When applying the simplified and non-simplified tracking models, we find they can be
improved slightly. After improvement, the correlation in regression analysis can be increased
significantly. We change Eqs. (1a) and (1b) to:
$$\ln c_{i+1} = a_1 + a_2 \psi + a_3 \lambda + a_4 \ln c_i + a_5 \theta_i + \varepsilon_c,\qquad\qquad(4a)$$
$$\theta_{i+1} = b_1 + b_2 \psi + b_3 \lambda + b_4 c_i + b_5 \theta_i + b_6 \theta_{i-1} + \varepsilon_\theta,\qquad\qquad(4b)$$
while the intensity model of Eq. (1c) remains unchanged. Accordingly, Eqs. (3a) and (3b) are
changed to:
$$\ln c_{i+1} = a_1 + a_2 \ln c_i + a_3 \theta_i + \varepsilon_c,\qquad\qquad(5a)$$




$$\theta_{i+1} = b_1 + b_2 c_i + b_3 \theta_i + \varepsilon_\theta, \qquad (5b)$$

while the intensity model of Eq. (3c) remains unchanged. Equations (1), (3), (4), and (5) are
named Model 1, Model 2, Model 3, and Model 4, respectively. Models 1 and 2 provide the
changes in $c$ and $\theta$ between times $i + 1$ and $i$, whereas in Models 3 and 4, we directly specify the
relationships between times $i + 1$ and $i$. That is, we directly calculate $c$ and $\theta$ at time-step $i + 1$
from time-step $i$, rather than calculate the changes between time steps $i + 1$ and $i$.
The fitting coefficient $a_i$ in Model 4 is illustrated in Fig. 1 from which we can observe its
spatial variation. Those for the other coefficients in Model 4 and the coefficients in Models 1–3
are not shown because of space limitations.

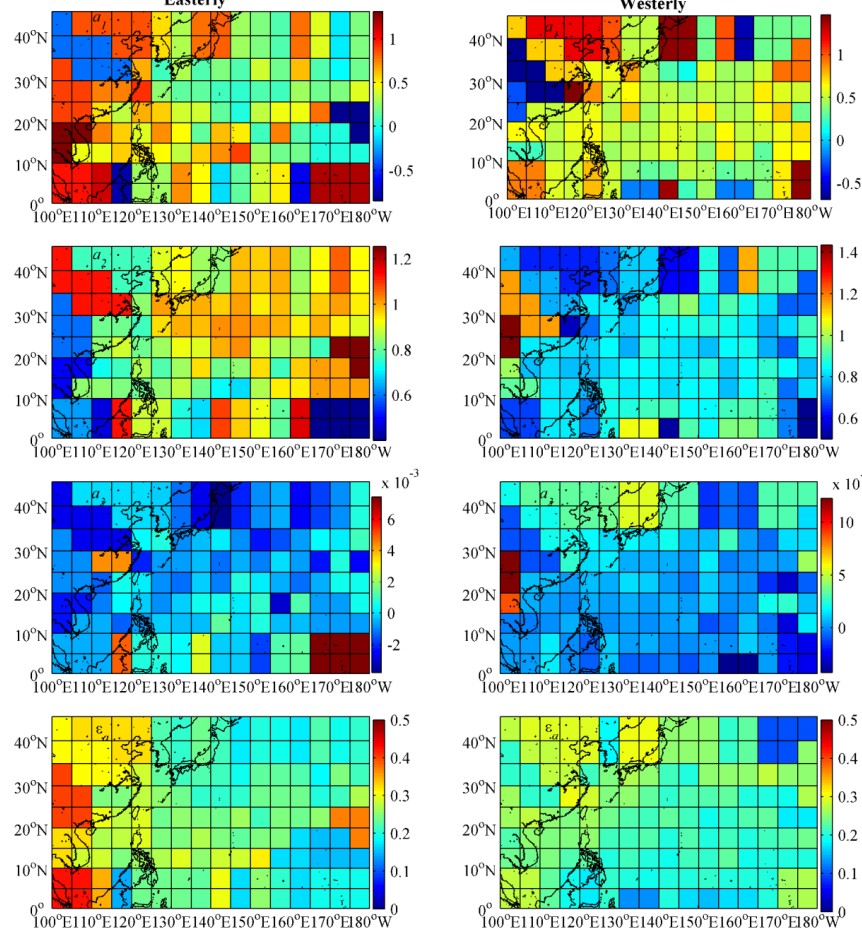


**Fig.1.** Illustration of regression coefficients $a$ in Model 4 for (left) easterly and (right) westerly headed storms.

We calculate the proportion of grid cells with correlation coefficient ($R^2$) >0.5 or >0.8 in all





grid cells for each coefficient's regression analysis in Models 1–4, and the results are shown in
Table 1. Comparison of Model 1 (Model 2) with Model 3 (Model 4) reveals that in the improved
tracking model, the proportions of grid cells with an $R^2$ value >0.5 and >0.8 are increased
significantly, which indicates the improved tracking model can improve the correlation in
regression analysis. The correlation coefficient ($R^2$) of each grid cell for fitting of the easterly and
westerly coefficient $a$ in Models 1 and 3 is shown in Figs. 2 and 3. It can be seen that the $R^2$ value
of each grid cell in Model 3 is significantly higher than in Model 1. Those for coefficient $b$ in
Models 1 and 3 and coefficients $a$ and $b$ in Models 2 and 4 are not shown because of space
limitations. It can also be seen that the $R^2$ value of each grid cell in Model 4 is significantly higher
than in Model 2. Comparison of Model 1 with Model 2 (Table 1) reveals that the $R^2$ values in both
models are reasonably low, and that the $R^2$ values of the simplified tracking model are slightly
lower than the non-simplified tracking model.

**Table 1.** Proportion of grid cells with correlation coefficient ($R^2$) greater than 0.5 or 0.8 in all grid cells for each
coefficient's regression analysis in Models 1–4. Largest value of $R^2$ for each coefficient is shown in bold.

| Model | Coefficient | Correlation coefficient | Proportion of grid cells | |
|---|---|---|---|---|
| | | | Easterly (%) | Westerly (%) |
| Model 1 | $a$ | $R^2 \geq 0.5$ | 15.97 | 9.72 |
| | | $R^2 \geq 0.8$ | 7.64 | 0 |
| | $b$ | $R^2 \geq 0.5$ | 27.08 | 15.97 |
| | | $R^2 \geq 0.8$ | 18.75 | 3.47 |
| Model 3 | $a$ | $R^2 \geq 0.5$ | **97.22** | **99.31** |
| | | $R^2 \geq 0.8$ | **47.22** | **27.78** |
| | $b$ | $R^2 \geq 0.5$ | **84.72** | **100** |
| | | $R^2 \geq 0.8$ | **33.33** | **31.94** |
| Model 2 | $a$ | $R^2 \geq 0.5$ | 6.25 | 2.08 |
| | | $R^2 \geq 0.8$ | 0 | 0 |
| | $b$ | $R^2 \geq 0.5$ | 12.50 | 11.11 |
| | | $R^2 \geq 0.8$ | 9.72 | 0 |
| Model 4 | $a$ | $R^2 \geq 0.5$ | **88.89** | **97.92** |
| | | $R^2 \geq 0.8$ | **40.28** | **26.39** |
| | $b$ | $R^2 \geq 0.5$ | **72.22** | **93.06** |
| | | $R^2 \geq 0.8$ | **20.14** | **23.61** |






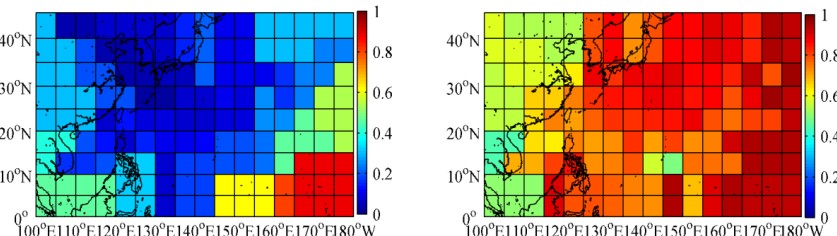

**Fig.2.** Correlation coefficient ($R^2$) of each grid cell for fitting of the easterly coefficient $a$ in Model 1 (left) and Model 3 (right).

**Fig.3.** Correlation coefficient ($R^2$) of each grid cell for fitting of the westerly coefficient $a$ in Model 1 (left) and Model 3 (right).

We also calculate the $R^2$ value when fitting coefficient $d$ in Models 1 and 2. Figure 4 shows the $R^2$ value of each grid cell for fitting of the easterly and westerly coefficient $d$ in Model 1, which shows the $R^2$ values of all grid cells are >0.8. Figure 5 shows the $R^2$ value of each grid cell for fitting of the easterly and westerly coefficient $d$ in Model 2. The $R^2$ value of 98.61% (97.92%) of grid cells is >0.8 for easterly (westerly) headed typhoons. From the above analysis, we find that the correlation for fitting of coefficient $d$ in Models 1 and 2 is generally better than for coefficients $a$ and $b$. This might be because the intensity model gives the statistical relationship between times $i + 1$ and $i$, which is similar to the improved tracking model, rather than the statistical relationship of changes between times $i + 1$ and $i$.

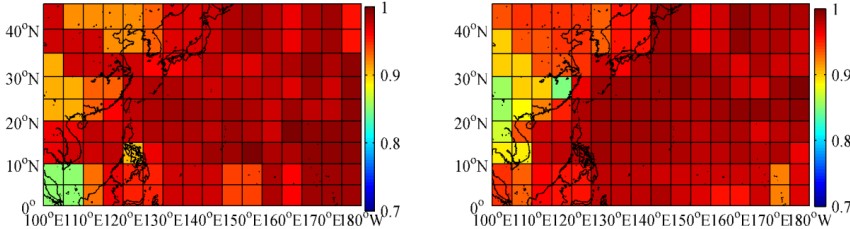

**Fig.4.** $R^2$ value of each grid cell for fitting of the easterly (left) and westerly (right) coefficient $d$ in Model 1.



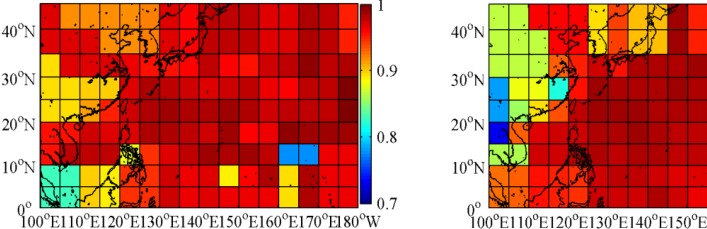


**Fig.5.** $R^2$ value of each grid cell for fitting of the easterly (left) and westerly (right) coefficient $d$ in Model 2.


**2.2 Validation of empirical track model**

Before using the empirical track model, we need to validate its efficiency. Section 2.1 showed

the correlation in regression analysis for Models 3 and 4 is better than for Models 1 and 2.
Therefore, we believe the improved models (Models 3 and 4) are superior to the original models
(Models 1 and 2). In the following, we consider only Models 3 and 4; therefore, only Models 3
and 4 are validated here.

Virtual typhoon events over 1000 years in the Northwest Pacific Ocean are simulated using

Models 3 and 4. The historical typhoon data used for verification were obtained from the China
Meteorological Administration dataset. Overall, 46 coastal stations are selected along the coast of
China, as shown in Fig. 6 (blue squares). Then, the typhoon events affecting each station (i.e.,
typhoons that pass within 250 km) are extracted from the virtual and historical typhoons datasets.
The use of a 250 km subregion has been suggested by Li and Hong (2015b, 2016) and by Vickery
et al. (2009a) following parametric investigation. Next, statistics such as mean annual occurrence
rate, the mean and standard central pressure difference, minimum approach distance, translation
speed, and storm heading are obtained for the simulated and historical tracks. All the values of
these key parameters (except the central pressure difference) are obtained when they are closest to
the coastal station. The central pressure difference is estimated using the minimum values within
the 250 km subregion. When a typhoon passes to the right (left) of a site, the minimum approach
distance is considered positive (negative).





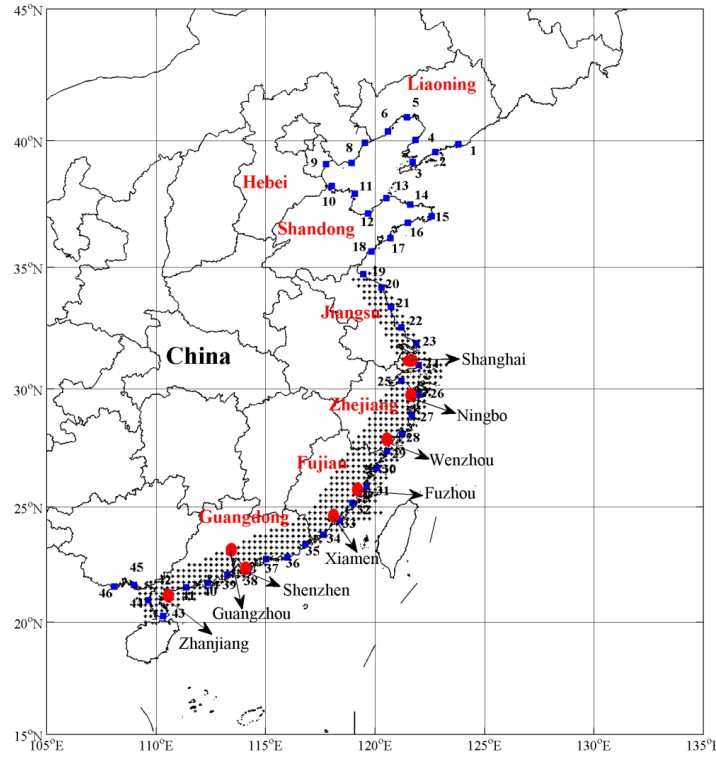


**Fig.6.** Locations of coastal stations (blue squares) along China's coastline, research points (black asterisks), and

coastal cities (red dots). Note: red text shows province names.


Figure 7 compares key parameters of typhoons simulated by Model 3 and observed typhoons

along China's coastline. The figure shows that the characteristics of simulated typhoons are in
good agreement with those from the observational dataset, which indicates that Model 3 can
reproduce the characteristics of typhoons along China's coastline.


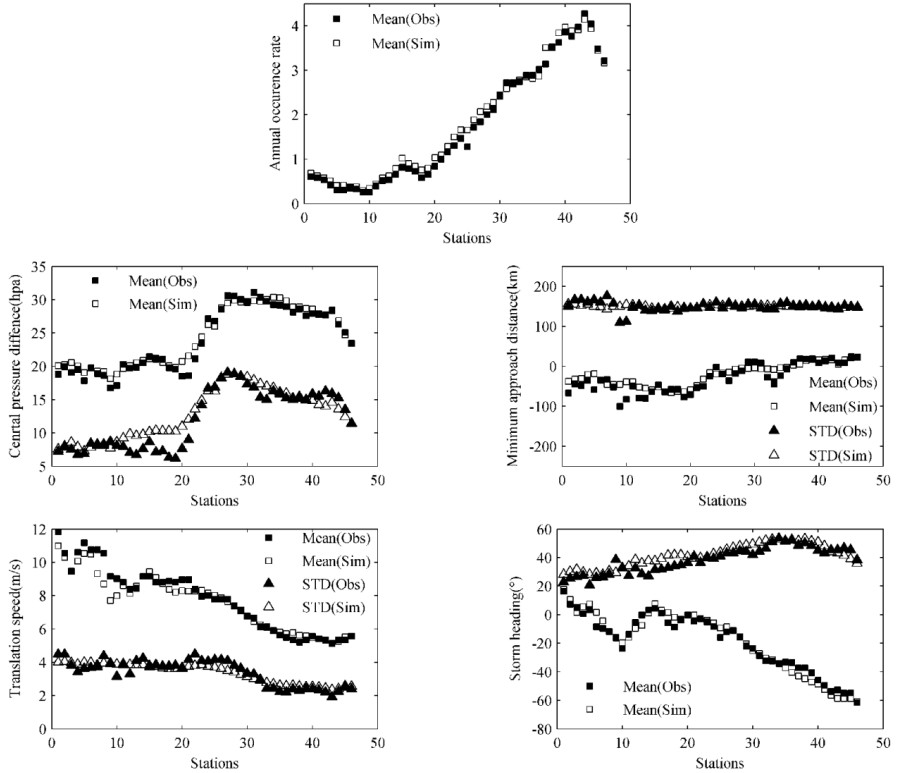

**Fig.7.** Comparison of key parameters of Model 3 simulated (Sim) and observed (Obs) typhoons at 46 coastal stations along China's coastline.

Figure 8 compares key parameters of typhoons simulated by Model 4 and observed typhoons along China's coastline. The figure shows that the characteristics of simulated typhoons also match well with those from the observational dataset, which indicates the performance of simplified Model 4 is comparable with non-simplified Model 3.





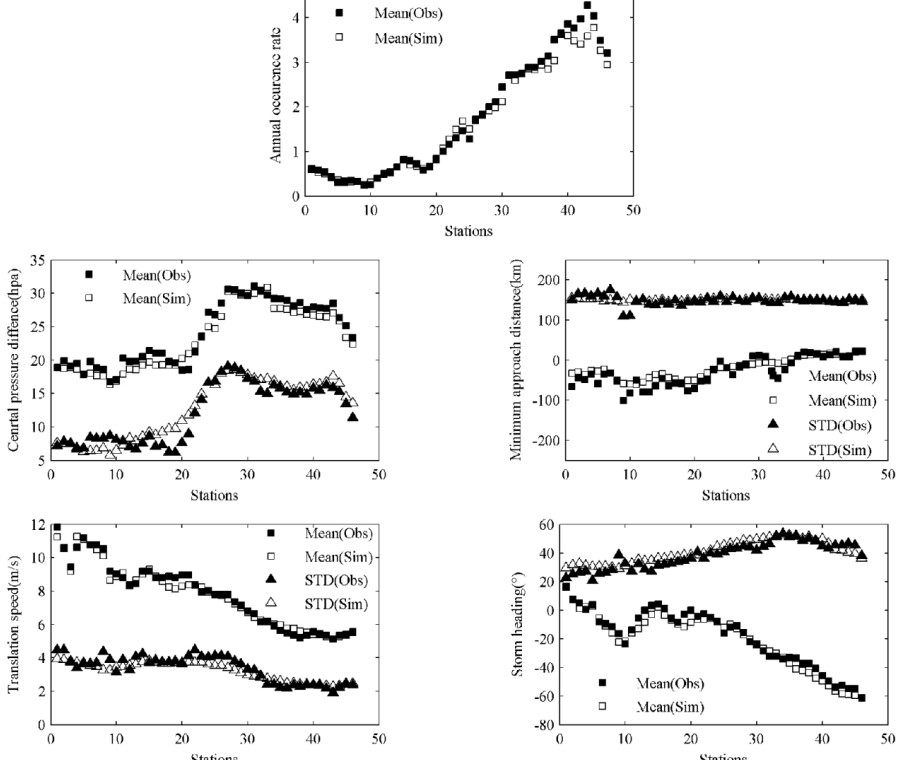

**Fig.8.** Comparison of key parameters of Model 4 simulated (Sim) and observed (Obs) typhoons at 46 coastal

stations along China's coastline.

## 3 Sensitivity of typhoon wind hazard model

The empirical track model is mainly used to generate large numbers of virtual typhoons to analyze the typhoon hazard. First, large numbers of virtual typhoons are obtained using the empirical track model and the decay model. Then, a research site is selected and the typhoon events that affect that site (i.e., those typhoons that pass within 250 km) are extracted from the virtual typhoons. Next, the wind field model is applied to calculate the wind speed (representing 10 min mean wind speed at 10 m height above the surface) of the extracted typhoons, from which samples of maximum wind speed are obtained. Finally, the samples are fitted by some extreme value distribution and the extreme wind speeds for different return periods are predicted. Many factors can influence the prediction of extreme wind speed throughout the entire process, e.g., different typhoon tracking models, different typhoon decay models, different statistical models for $R_{max}$ and $B$, and different extreme value distributions.

To explore the sensitivity of the typhoon wind hazard model to the above four factors, we





calculate the extreme wind speeds for different return periods under the influence of different
factors and make a comparison. To map the typhoon wind hazard, we select 579 grid points as
research sites in the southeast coastal region of China, as shown in Fig. 6 (black asterisks). The
grid resolution is set to 0.25°, and for each research site, the extreme wind speeds at 50- and
100-year return periods are predicted under the influence of the different factors.
The Yan Meng (YM) wind field model, developed by Meng et al. (1995), is applied in this
study to calculate the wind speed. As indicated by Meng et al. (1995), the model involves moving
wind field model of typhoons and introduces the concept of the "equivalent roughness length" to
consider topographical effects. The YM model is sufficiently accurate for typhoon simulation and
it has been applied by Matsui et al. (2002), Okazaki et al. (2005), and Xie et al. (2015). For
additional details regarding the wind field model, the reader is referred to Meng et al. (1995). The
wind speed calculated by the YM model is an hourly mean and the ratio of the maximum 10 min
mean wind speed to the hourly mean is equal to 1.06.

**3.1 Influence of different decay models on extreme wind speeds**
When a typhoon makes landfall, its intensity will weaken because of the loss of energy from
the sea and because of increased ground friction. Modeling the decay of typhoons after landfall
plays an important role in typhoon hazard analysis at coastal stations. We first investigate the
influence of the typhoon decay model on predicted wind speed. Model 3 is used to generate virtual
typhoon events in the Northwest Pacific Ocean, and in this process, we apply two different decay
models. One is the model developed by Vickery and Twisdale (1995b):

$$\Delta p(t) = \Delta p_0 \exp(-at); \quad a = a_0 + a_1 \Delta p_0 + \varepsilon \quad , \tag{6}$$

where $\Delta p(t)$ is the central pressure difference (hPa) at time $t$ after landfall, $\Delta p_0$ is the central
pressure difference (hPa) at landfall, $a$ is the decay constant, and $\varepsilon$ is a normally distributed error
term. The other model is the model developed by Vickery (2005):

$$\Delta p(t) = \Delta p_0 \exp(-at); \quad a = a_0 + a_1 \Delta p_0 c / R_{\max} + \varepsilon , \tag{7}$$

where $c$ is the typhoon translation speed at landfall (km h$^{-1}$), and $R_{\max}$ is the radius to maximum
winds at landfall (km). Vickery (2005) indicated that Eq. (7) can increase the correlation
coefficient R$^2$ in regression analysis (coefficients $a_0$ and $a_1$ are determined by regression analysis)
on the Gulf Coast, Florida Peninsula, and Atlantic Coast of the USA.
The typhoon landing area in the Northwest Pacific Ocean is divided into five subregions: the
region north of 30°N (extratropical cyclone area, Zone1), region between 25°N and 30°N (area
north of Taiwan, Zone2), region between 20°N and 25°N (area including Taiwan, Zone3), region
of The Philippine Islands (Zone5), and region of the remaining areas (Zone4). The fitting




coefficients of Eqs. (6) and (7) are summarized in Table 2, where $N$ is the number of data points
used for the regression analysis, $R^2$ is the correlation coefficient, and $\sigma_\varepsilon$ is the standard deviation
of the errors. In Table 2, the largest value of $R^2$ is shown in bold for each region examined. It can
be seen that the correlation in the decay model of Vickery and Twisdale (1995b) is better than that
of Vickery (2005) for most regions.

**Table 2.** Decay constant $a$ in Eqs. (6) and (7). Numbers in bold type are the largest $R^2$ value for each region.

| Region | $N$ | $a = a_0 + a_1 \Delta p_0 + \varepsilon$ | | | | $a = a_0 + a_1 \Delta p_0 c / R_{max} + \varepsilon$ | | | |
|--------|-----|------|------|------|------|------|------|------|------|
| | | $a_0$ | $a_1$ | $R^2$ | $\sigma_\varepsilon$ | $a_0$ | $a_1$ | $R^2$ | $\sigma_\varepsilon$ |
| Zone1 | 36 | 0.0078 | 0.00075 | **0.0928** | 0.0198 | 0.0293 | 0.00004 | 0.00018 | 0.0194 |
| Zone2 | 66 | 0.0161 | 0.00055 | **0.0946** | 0.0203 | 0.0244 | 0.00049 | 0.0589 | 0.0207 |
| Zone3 | 159 | 0.0137 | 0.0012 | 0.2139 | 0.0247 | 0.0291 | 0.0011 | **0.2157** | 0.0242 |
| Zone4 | 82 | -0.0035 | 0.0019 | **0.4768** | 0.0216 | 0.0101 | 0.0020 | 0.4565 | 0.0220 |
| Zone5 | 40 | -0.0026 | 0.00052 | **0.5321** | 0.0116 | -0.00006 | 0.00078 | 0.4374 | 0.0127 |


In Sect. 2, we described the use of Model 3 and the decay model of Vickery and Twisdale
(1995b) to generate virtual typhoons and to validate their statistical characteristics. Here, we use
Model 3 in combination with the new decay model of Vickery (2005) to generate virtual typhoons
for the Northwest Pacific Ocean and to validate its efficiency. Because of space limitations, the
results of the verification are not given here. The numerical experiment using Model 3 and Eq. (6)
to predict the wind speed is referred to as Test 1, and that using Model 3 and Eq. (7) is referred to
as Test 2. In Tests 1 and 2, $R_{max}$ and $B$ are calculated based on the models given in Vickery and
Wadhera (2008):
$$\ln R_{max} = 3.015 - 6.291 \times 10^{-5} \Delta p^2 + 0.0337 \psi + \varepsilon_{\ln R max}, \quad B = 1.833 - 0.326 \sqrt{1000 f_c R_{max}} + \varepsilon_B, \quad (8)$$

where $\Delta p$ is in hPa; the standard deviation of $\varepsilon_{\ln R max}$, $\sigma_{\ln R max} = 0.448$ for $\Delta p \leq 87$ hPa, $1.137 -$
$0.00792 \Delta p$ for $87$ hPa $< \Delta p \leq 120$ hPa, and $0.186$ for $\Delta p > 120$ hPa; $\psi$ is latitude (°); $f_c$ is the
Coriolis parameter; and $\sigma_B = 0.221$.
The empirical distribution is used as the extreme value distribution in both Test 1 and Test 2.
Table 3 shows the settings for Tests 1 and 2 as well as other tests described in the following
section of this paper.

**Table 3.** Settings for different tests (those in the same color represent a set of controlled trials).

| Test | Decay model | Track model | $R_{max}$ and $B$ model | Extreme value distribution |
|------|-------------|-------------|--------------------------|----------------------------|
| Test 1 | Eq.(6) | Model 3 | Eq. (8) | Empirical |
| Test 2 | Eq.(7) | Model 3 | Eq. (8) | Empirical |




| Test 3 | Eq.(6) | Model 4 | Eq. (8) | Empirical |
| Test 4 | Eq.(6) | Model 3 | Eq. (9) | Empirical |
| Test 5 | Eq.(6) | Model 3 | Eq. (10) | Empirical |
| Test 6 | Eq.(6) | Model 3 | Eq. (8) | Weibull |
| Test 7 | Eq.(6) | Model 3 | Eq. (8) | Gumbel |
| Test 8 | Eq.(6) | Model 3 | Eq. (8) | GPD |


The predicted extreme wind speeds for a 50-year return period ($V_{50}$) for 579 stations in the
southeast coastal region of China are used to map the typhoon hazard, as shown in Fig. 9. The
results predicted by Tests 1 and 2 are shown in Fig. 9(a) and (b), respectively. It can be seen from
Fig. 9 that the different decay models, i.e., Eqs. (6) and (7), have little impact on the predicted
wind speed, and that the maximum difference (MD) of wind speed is only about 0.5 m s$^{-1}$. We
also compare the predicted wind speeds for a 100-year return period ($V_{100}$) for Tests 1 and 2 (not
shown because of space limitations). The MD is also about 0.5 m s$^{-1}$ and the maximum relative
difference (MRD) is only about 1%.

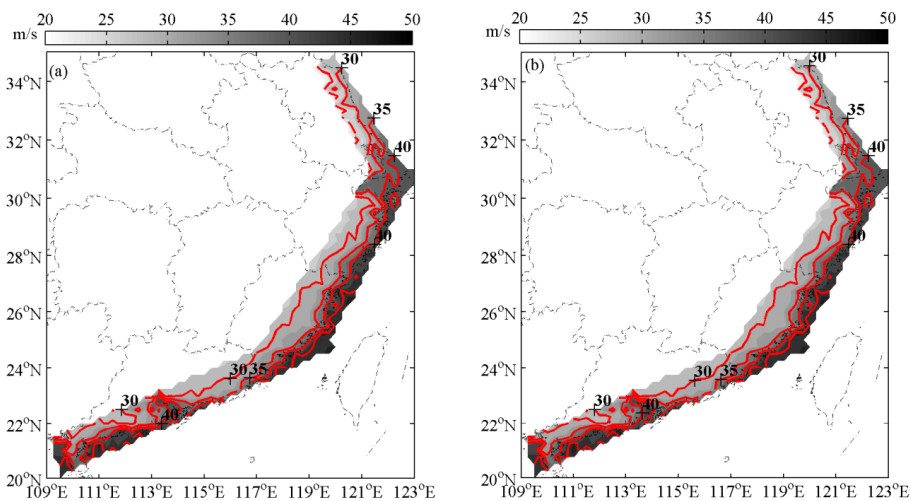

**Fig.9.** Maps of extreme wind speeds (m/s) for 50-year return period in (a) Test 1 and (b) Test 2.

**3.2 Influence of different track models on extreme wind speeds**
In Sect. 2, the non-simplified and simplified track models (Models 1 and 2) are improved to
produce Models 3 and 4, and we validate the virtual typhoons generated using Models 3 and 4. To
investigate the influence of the non-simplified and simplified track models on predicted extreme
wind speeds, we estimate $V_{50}$ and $V_{100}$ for China's southeast coast based on the virtual typhoons
generated using Models 3 and 4. In this process, the decay model of Eq. (6), $R_{max}$ and $B$ model of





Eq. (8), and the empirical distribution are adopted. The numerical experiments are referred to as
Tests 1 and 3, as shown in Table 3. The predicted $V_{50}$ in Test 1 is shown in Fig. 9(a). The
estimated $V_{50}$ in Test 3 is shown in Fig. 10(a) and the wind speed difference between Tests 1 and 3
is shown in Fig. 10(b). It can be seen from Fig. 10(b) that the wind speeds predicted by the
non-simplified track model (Test 1) are larger than predicted by the simplified track model (Test 3)
on most of the southeast coast of China, especially in the coastal regions of Zhejiang and Fujian
provinces. The MD of predicted wind speed is about 3.5 m s$^{-1}$ and the MRD is about 10%. For the
estimated $V_{100}$, there is a similar spatial trend; the MD is about 4.5 m s$^{-1}$ and the MRD is about

12%.

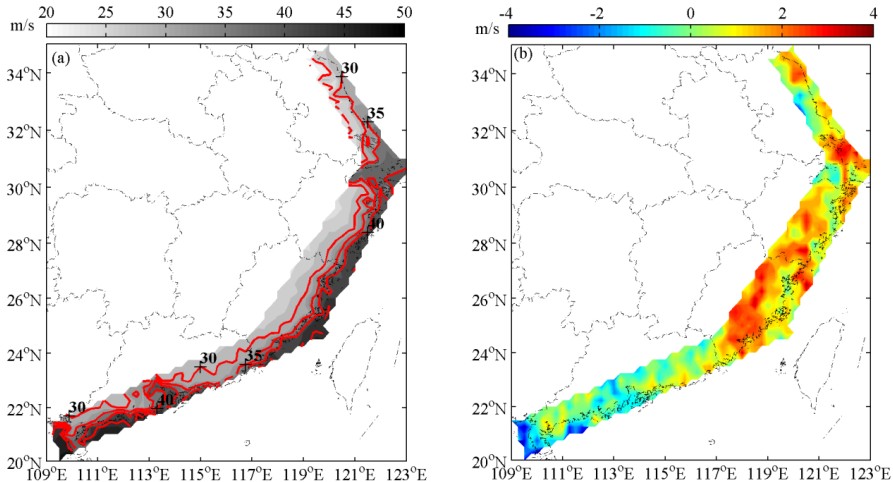


**Fig.10.** Maps of extreme wind speeds (m/s) for 50-year return period in (a) Test 3 and (b) the wind speed
difference (m/s) between Tests 1 and 3.

**3.3 Influence of different $R_{max}$ and $B$ models on extreme wind speeds**
In the typhoon wind field model, $R_{max}$ and $B$ are important parameters. Their calculation
formulas influence the wind speed calculated by the wind field model, which subsequently
influences the prediction of extreme wind speed. We select three different models to investigate
the influence of $R_{max}$ and $B$ on the predicted wind speed. One is the model developed by Vickery
and Wadhera (2008), as mentioned in Sect. 3.1. This model has been used by Li and Hong (2015a,
2015b, and 2016) and by Hong et al. (2016). The second model was developed by Vickery et al.
(2000) and it has been used by Pei et al. (2014). The model can be expressed as follows:
$$\ln R_{max} = 2.636 - 0.0000508\Delta p^2 + 0.0394\psi; \quad B = 1.38 + 0.00184\Delta p - 0.00309 R_{max}. \quad (9)$$
The third model was developed by Xiao et al. (2011) based on the typhoons that affect China's
coast region and some empirical information from other literature. The model can be expressed as





follows:

$$\ln R_{\max} = c_0 + c_1 \Delta p + \varepsilon_1; \quad \ln B = d_0 + d_1 \ln R_{\max} + \varepsilon_2 \quad , \tag{10}$$

where $c_0$, $c_1$, $d_0$, and $d_1$ are model coefficients and $\varepsilon_1$ and $\varepsilon_2$ are normally distributed error terms
with mean zero. For values of these parameters and the standard deviations of $\varepsilon_1$ and $\varepsilon_2$, the reader
is referred to Xiao et al. (2011).
We compare $R_{\max}$ and $B$ calculated by the three models with latitude $\psi$ set to 25 °N. The
comparison results are shown in Fig. 11. It can be seen that when $\Delta p$ is <60 hpa, the mean of $R_{\max}$
calculated by Eq. (10) is larger than calculated by Eqs. (8) and (9), and when $\Delta p$ is >60 hpa, the
mean of $R_{\max}$ calculated by Eq. (10) is slightly smaller than calculated by Eqs. (8) and (9). The
mean of $B$ estimated by Eq. (10) is much greater than predicted by Eqs. (8) and (9), although the $B$
value is within the range suggested by Willoughby and Rahn (2004), Vickery et al. (2000), and
Holland (1980). Both $R_{\max}$ and $B$ calculated by Eqs. (8) and (9) have little difference. The mean of
$R_{\max}$ calculated by Eq. (8) is slightly greater than calculated by Eq. (9), while the mean of $B$
calculated by Eq. (8) is slightly smaller than calculated by Eq. (9).

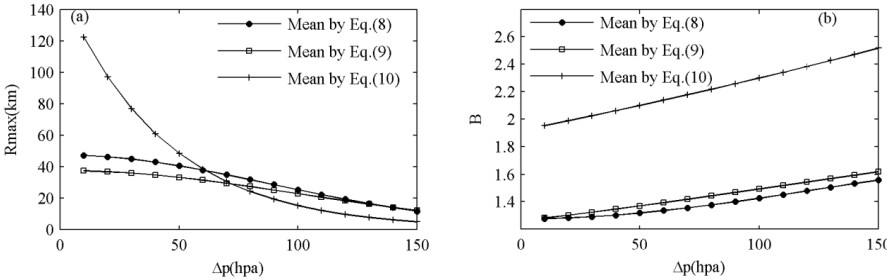


**Fig.11.** Comparison of estimated (a) $R_{\max}$ and (b) $B$ using Eqs. (8), (9), and (10).

In Test 1, Model 3 combined with the decay model of Eq. (6), the $R_{\max}$ and $B$ model of Eq. (8)
and the empirical distribution are used to predict the wind speed for different return periods. Here,
we use the different $R_{\max}$ and $B$ models (Eqs. (9) and (10)) to predict the wind speed, named as
Test 4 and Test 5. The specific settings for Tests 1, 4, and 5 are shown in Table 3. Figure 12
shows the estimated $V_{50}$ in Test 4 (Fig. 12(a)) and the wind speed difference between Tests 4 and 1
(Fig. 12(b)). It can be seen from Fig. 12(b) that Test 1 underestimates wind speed in comparison
with Test 4 in coastal regions of Jiangsu, Zhejiang, and Fujian provinces. The MD of the predicted
wind speed is about 2 m s$^{-1}$ and the MRD is about 5%. This should be because the $B$ value
calculated by Eq. (9) is slightly larger than calculated by Eq. (8). In coastal regions of Guangdong
Province, the estimated $V_{50}$ in Test 4 is slightly larger but it has little difference from that in Test 1.
This might be because the $\Delta p$ along the coast of Guangdong Province increases significantly (see
Figs. 7 and 8) and the difference of $R_{\max}$ calculated by Eqs. (8) and (9) decreases according to Fig.



11(a), leading to the smaller difference of the predicted wind speed. For the estimated $V_{100}$, there
is a similar spatial trend; the MD is about 2.8 m s$^{-1}$ and the MRD is about 7%.

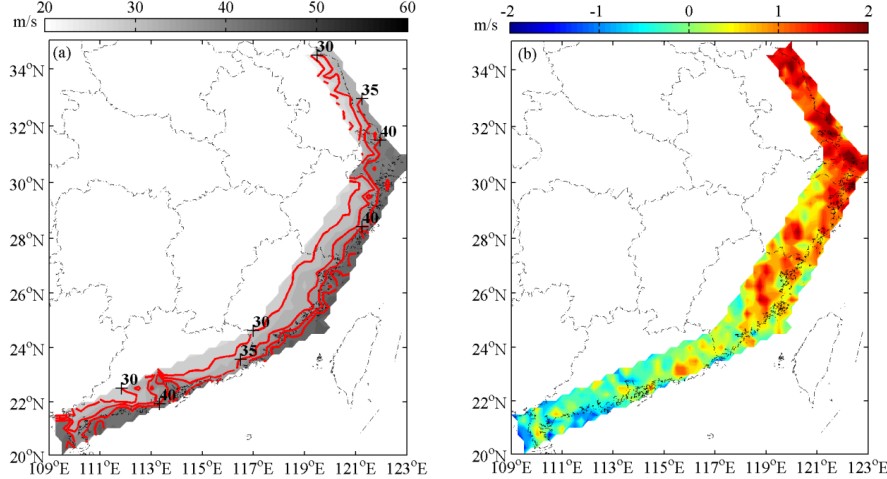

**Fig.12.** Maps of extreme wind speed (m/s) for 50-year return period in (a) Test 4 and (b) the wind speed difference

(m/s) between Tests 4 and 1.

Figure 13 shows the estimated $V_{50}$ in Test 5 (Fig. 13(a)) and the wind speed difference
between Tests 5 and 1 (Fig. 13(b)). It can be seen from Fig. 13(b) that the wind speed predicted by
Test 5 is significantly higher than predicted by Test 1 throughout the entire southeast coastal
region of China. The MD of the predicted wind speed is up to 15 m s$^{-1}$ and the MRD is about 37%.
This is because the $B$ value calculated by Eq. (10) is significantly greater than calculated by Eq.
(8). For the estimated $V_{100}$, the MD increases to 21 m s$^{-1}$ and the MRD is about 50%.

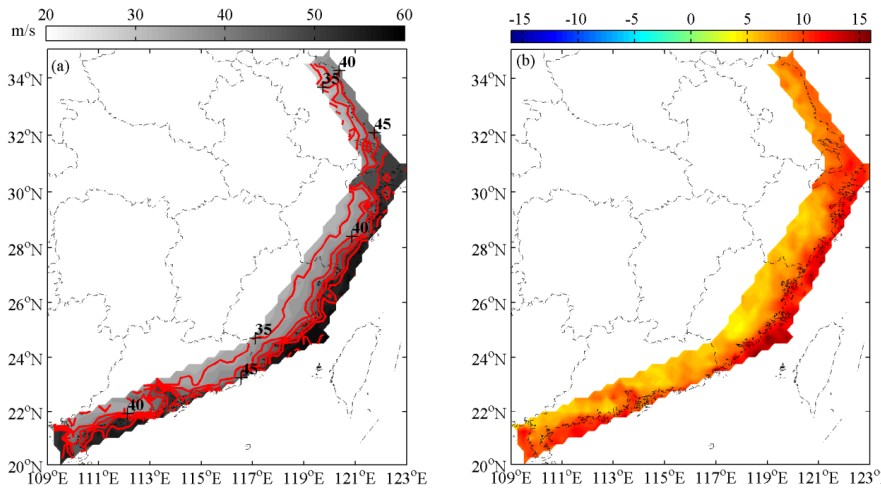

**Fig.13.** Maps of extreme wind speed (m/s) for 50-year return period in (a) Test 5 and (b) the wind speed difference



(m/s) between Tests 5 and 1.


**3.4 Influence of different extreme value models on extreme wind speeds**

The samples of maximum wind speed obtained through numerical simulation need to be

fitted by some extreme value distribution to predict the extreme wind speed of different return
periods. In typhoon hazard analysis, the commonly used extreme value distributions include
Extreme-I distribution (i.e., the Gumbel distribution), Extreme-II distribution (i.e., the Frechet
distribution), and Extreme-III distribution (i.e., the Weibull distribution). If the sample size is
sufficiently large, the empirical distribution should be preferred because there is no assumption
about the tail shape of the wind speed distribution. The sample of maximum wind speed is initially
considered to obey the Extreme-II distribution (Thom, 1960). However, more studies have shown
that the Extreme-I distribution is more suitable (Simiu et al. 1980; Simiu and Filliben, 1976). In
recent years, some studies have found that the peaks-over-threshold method with the generalized
Pareto distribution (GPD) can provide satisfactory wind speed estimation (Simiu and Heckert,
1995). Different extreme value distributions will have impact on the predicted extreme wind speed.
In this study, we apply the empirical distribution, Weibull distribution, Gumbel distribution, and
GPD to explore the influence of these four different distributions on the prediction of extreme
wind speed.
The Weibull distribution takes the form

$$F_W(x) = 1 - \exp\left[-\left(\frac{x-\gamma}{\eta}\right)^{\beta}\right]. \qquad (11)$$

The Gumbel distribution takes the form

$$F_G(x) = \exp\left\{-\exp\left[-\left(\frac{x-\gamma}{\eta}\right)\right]\right\}. \qquad (12)$$

The GPD function is as follows

$$G(x) = 1 - (1 + \beta\frac{x-u}{\eta})^{-\frac{1}{\beta}}. \qquad (13)$$

where $x$ is the corresponding variable; $\gamma$, $\eta$, $\beta$ is the position parameter, scale parameter and shape
parameter, respectively; $u$ is the threshold value.

In Test 1, the empirical distribution is adopted. Taking Test 1 as the controlled trial, the

numerical experiments adopting the Weibull distribution, Gumbel distribution, and GPD are
defined as Test 6, Test 7, and Test 8, respectively. The specific settings for Tests 1 and 6–8 are
listed in Table 3.

Figure 14 shows the estimated $V_{50}$ in Test 6 (Fig. 14(a)) and the wind speed difference

between Tests 6 and 1 (Fig. 14(b)). It can be seen from Fig. 14(b) that in most areas of China's





southeast coasts, the wind speed predicted by the Weibull distribution is lower than predicted by the empirical distribution, especially in Fujian Province. The MD of the predicted wind speed is about $-3$ m s$^{-1}$ and the MRD is about 7%. For the estimated $V_{100}$, the MD is about $-4$ m s$^{-1}$ and the MRD is about 10%.

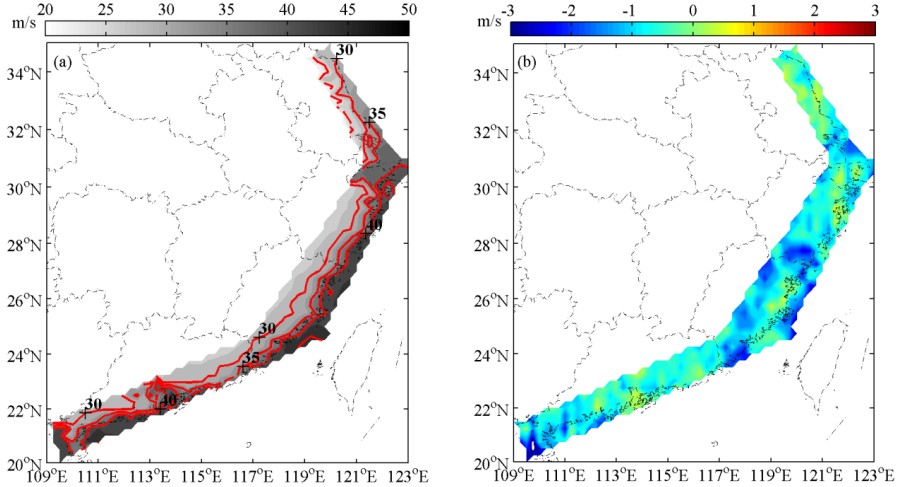

**Fig.14.** Maps of extreme wind speed (m/s) for 50-year return period in (a) Test 6 and (b) the wind speed difference (m/s) between Tests 6 and 1.

Figure 15 shows the estimated $V_{50}$ in Test 7 (Fig. 15(a)) and the wind speed difference between Tests 7 and 1 (Fig. 15(b)). Figure 15(b) indicates that over the entire southeast coastal region of China, the wind speed predicted by the Gumbel distribution is higher than predicted by the empirical distribution, especially in Guangdong Province. The MD of the predicted wind speed is about 8 m s$^{-1}$ and the MRD is about 20%. For the estimated $V_{100}$, the MD increases to 10 m s$^{-1}$ and the MRD is about 25%.

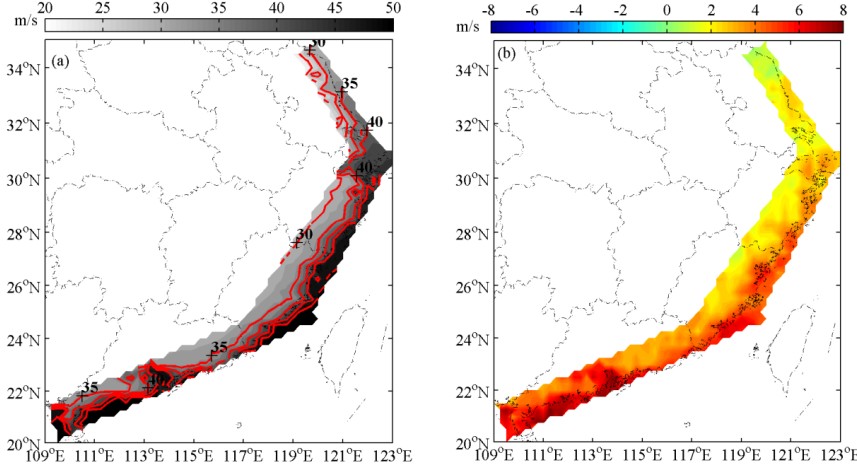





**Fig.15.** Maps of extreme wind speed (m/s) for 50-year return period in (a) Test 7 and (b) the wind speed difference
(m/s) between Tests 7 and 1.


Figure 16 shows the estimated $V_{50}$ in Test 8 (Fig. 16(a)) and the wind speed difference between
Tests 8 and 1 (Fig. 16(b)). Figure 16(b) shows that over the entire southeast coastal region of
China, the wind speed predicted by the GPD is lower than predicted by the empirical distribution,
especially in Fujian and Guangdong provinces. The MD of the predicted wind speed is about −7 m
s$^{-1}$ and the MRD is about 17%. For the estimated $V_{100}$, the MD is about −8 m s$^{-1}$ and the MRD is
about 20%.

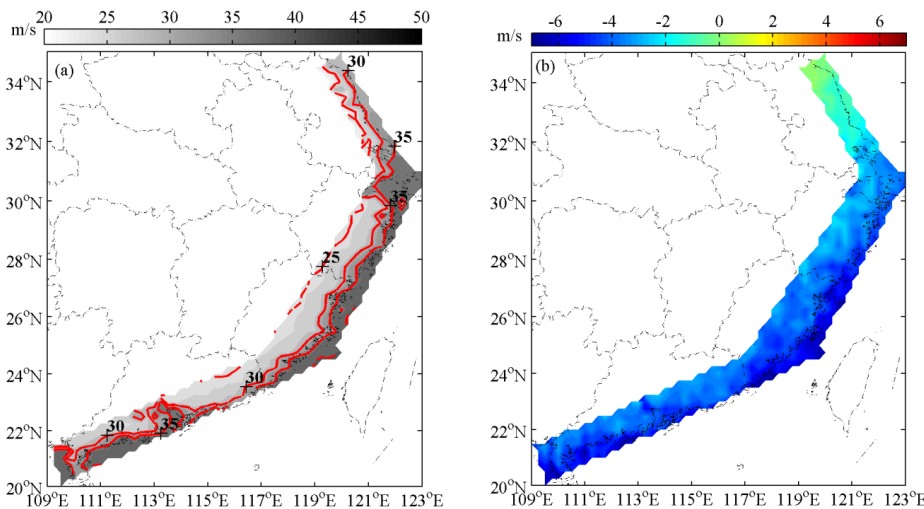


**Fig.16.** Maps of extreme wind speed (m/s) for 50-year return period in (a) Test 8 and (b) the wind speed difference
(m/s) between Tests 8 and 1.


**3.5 Estimation of typhoon wind hazard for eight cities**

In addition to the typhoon hazard analysis conducted for the southeast coastal region of China,

we also estimate the typhoon wind hazard for eight key coastal cities of China under the influence
of different factors and we compare the results with the Chinese design code (GB 50009, 2012).
For details of the design code values of 50-year and 100-year return periods for these cities, the
reader is referred to Li and Hong (2016). Figure 17 shows the $V_{50}$ (Fig. 17(a)) and $V_{100}$ (Fig. 17(b))
of the eight cities predicted by Tests 1–8 and the values from the code. For most cities, it can be
seen that the wind speed predicted by Test 1 is consistent with the code except for Wenzhou,
which indirectly proves the reliability of the method used in this paper to predict the extreme wind
speed. For Wenzhou, Test 1 overestimates wind speed by about 15% in comparison with the code.
The extreme wind speed predicted by Tests 1–4 and Test 6 have little difference, i.e., the relative
difference is within 10%.





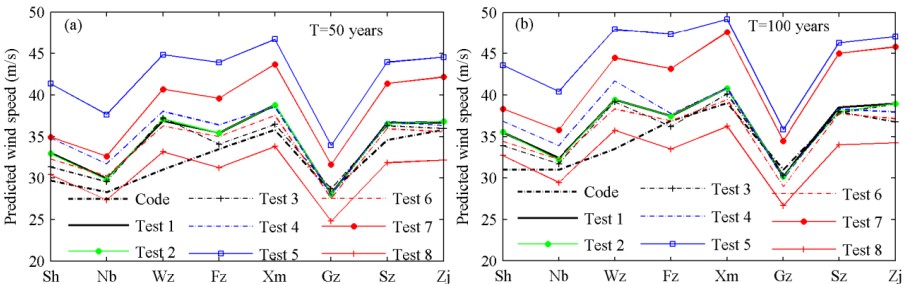

**Fig.17.** (a) $V_{50}$ and (b) $V_{100}$ of the eight cities predicted using Tests 1–8 and the code. Shanghai (Sh), Ningbo (Nb), Wenzhou (Wz), Fuzhou (Fz), Xiamen (Xm), Guangzhou (Fz), Shenzhen (Sz), and Zhanjiang (Zj).

## 4 Conclusions

In this paper, we describe a technique for analyzing typhoon hazard based on the empirical track model. The existing simplified and non-simplified typhoon empirical track models are improved. In the improved tracking models, the correlation in regression analysis is increased significantly. We also quantitatively investigate the sensitivity of the typhoon wind hazard model to different typhoon decay models, the simplified and non-simplified typhoon tracking models, different statistical model for $R_{max}$ and $B$, and different extreme value distributions. We found the different typhoon decay models have least influence on the predicted extreme wind speed, and the MRD from the control group is only about 1%. Over most of the southeast coast of China, the predicted wind speed by the non-simplified typhoon tracking model is larger than from the simplified tracking model, especially in Zhejiang and Fujian provinces. The MRD of predicted wind speed for a 50-year return period ($V_{50}$) is about 10%. The use of different models of $R_{max}$ and $B$ has considerable impact on the predicted wind speed, and the MRD of $V_{50}$ can reach up to 37%. This depends mainly on the difference of the $B$ value calculated by the different models. Throughout the southeast coast of China, the predicted wind speed from the Weibull distribution is lower than from the empirical distribution, especially in Fujian Province. The MRD of the $V_{50}$ is about 7%. The predicted wind speed from the Gumbel distribution is higher than from the empirical distribution, especially in Guangdong Province, and the MRD for $V_{50}$ is up to 20%. The predicted wind speed from the GPD is lower than from the empirical distribution, especially in Fujian and Guangdong provinces, and the MRD for $V_{50}$ is up to 17%. For several coastal cities of China, the predicted wind speeds in this paper are consistent with those from the design code. This paper constitutes a useful reference for predicting extreme wind speed when using the empirical track model.

In this paper we improve the empirical track model and use it to analyze the typhoon hazard for southeast coastal region of China. This hazard model can overcome the problem that one can't estimate the typhoon wind speeds as a function of return period using the traditional methods,





because the lack of the measured wind-speed data. Besides we investigate the influence of different factors on the predicted wind speeds. This study's results could be valuable to 1) urban planners and emergency managers responsible for typhoon disaster preparedness, response, and recovery planning; 2) policy-makers to evaluate the adequacy of structural design codes, and 3) insurance companies to assess real properties and adjust typhoon hazard insurance rates.

The study of typhoon hazard risk includes the prediction of typhoon intensity and frequency and the study of typhoon wind speed for different return periods. Combining typhoon accurate forecast, typhoon speed estimation of different return periods with hazard loss assessment from natural, social, economic, policy, cultural and engineering perspectives, a comprehensive risk assessment framework and index system for typhoon hazard can be established. A comprehensive study on the tolerance and response mechanism of coastal cities to typhoon hazard will be the focus of our next work.

### Data availability statement

The observed typhoon data that support the findings of this study are available in the CMA-repository (http://tcdata.typhoon.org.cn). The datasets generated during the current study are available from the corresponding author on reasonable request.

### Acknowledgement

This study was funded by the National Key Research and Development Program of China (Grant No. 2016YFC1402004, 2016YFC1402000, 2018YFC1407003). Data from the CMA-STI Best Track Dataset for Tropical Cyclones over the Western North Pacific online dataset are gratefully acknowledged. Thanks are extended to reviewers.

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
