# Peer review of "Improvement of typhoon wind hazard model and its sensitivity 1"

_Natural Hazards and Earth System Sciences, 2018_

## Referee Comment (RC1) · Anonymous Referee #1 · 9 Sep 2019

Dear Dr. James Daniell, I have completed my review of this manuscript and provided my comments as follows. The main goal of this manuscript was to investigate the "sensitivity" of different model components in the tropical cyclone wind hazard modeling. There were four targets this paper was trying to establish for a scientific contribution. First exercise was to repeat a statistical track model proposed in previous study and worked out a different version. The second was to use different statistical models for B and Rmax from other studies. The third was to check the developed filling rate model. The last exercise was to fit the simulated wind speed into different probabilistic distribution. My general comment is that this paper failed to conduct creative works to achieve the goals it was trying to establish. Most of the context in this paper was repetitive work in essence and could not provide more insights. Section 2 in this paper was to repeat

a statistical model proposed in Li and Hong (2015) and tried to improve it. It should be noted that it has been noted in Li (2016) that different form of track model could be developed. The one that showed in this paper had been investigated in their study but not reported due to the physical limits could be easily beyond using this model. In other words, a direct modeling in translational velocity, heading, even central pressure/pressure difference (if one wants) could have better R2. If author tried, instead of log relation, the direct use of $c_i$ and $C_{i+1}$ could provide even better fitting. The reason why is that these values are calculated by consecutive storm position linearly. One obviously should except a very good linear relation at adjacent track point. However, the problem with this kind of fitting was its engineering implication. The simulation could generate storms that goes east at this step and suddenly west at the next step because of the total randomness. In other words, the claimed "improved" model in this paper could easily simulate event having no physical meaning. The goodness of fit can show the trend of different variables but should not be used as a single measurement to determine the improvement. These drawbacks were not discussed in this paper at all. If truncation or removal of these error simulated events had been conducted in their model but not reported in this paper, it would greatly mislead the general readers, who may not have sufficient experience in modeling works. Extensive validation has to be made to show this statistical chaos can be reasonably avoided if this proposed model was used. It is not clear that whether this contributes to the weird looking wind contours shown in Figure 14 to Figure 16. It should be also noted that the "simplified" model mentioned in Li and Hong (2015) was to compare Vickery et al. (2000), their study have shown that there is redundancy in the modeling process. If following the same idea of this paper, using a model fitting measure known as AIC, the model proposed in Li and Hong (2015) is better than Vickery et al. (2000) due to the reduced independent variables. However, no claims of improvement were made in their study due to the above reasons. In terms of the second exercises, I strong against the idea that considers the statistical model developed in different periods, which used the same data source but with different length of records, as uncertainty. It should be noted that

all available data to develop the Rmax and B was mainly in North Atlantic. The model developed in Vickery et al. (2000) used data earlier than 2000. However, the model developed in Vickery et al. (2008) used data much more completed than Vickery's 2000 model. In other words, the author was comparing an outdated model to the latest model and claim there was "uncertainties". This comparison is not appropriate at all. The whole associated sections become meaningless from this perspective. The paper also missed reviewed several existing publications that has developed filling rate model for mainland China, e.g., Li and Hong (2015) as mentioned in this paper. For the developed model in this paper, I recommended that the model needs to be disjoint from the Rmax model as this data is neither available nor very reliable at or after the storm making landfall. The last exercise used different probabilistic data to fit the simulated wind speed, which totally lose its point of developing long term simulation at all. In wind engineering practice, adopting a probabilistic distribution was a choice to deal with the limited data. One cannot predict the 100-year return period wind speeds with only 50 years of data without fitting it into a distribution, which extrapolate the data into a longer return period. However, the simulation could generate more than 100,000 years of data. Why one still need an approximated distribution than its parent true distribution from the data? None of the building code that has been adopted storm simulation still use fitting technique to get the return period. The probabilistic model is only needed for historical limited data. From the above points, I DO NOT consider this paper could be a contribution to those who have experience in tropical cyclone simulations. Oppositely, this paper might bias the general reader in terms of what is important in the modeling and what has been achieved in the modeling field. As mentioned above, the authors failed to establish evidences to show this study had any improvement than the existing publications. I recommend the authors should take more efforts in revising these statistical models and add more physical modeling into the engineering statistical modeling.

[Figure]

2018-390, 2019.

---

## Author Comment (AC1) · 10 Sep 2019

I think the reviewer is right. We need to reorganize and revise what we have done.